# Predictor of HPV Vaccination Uptake among Foreign-Born College Students in the U.S.: An Exploration of the Role of Acculturation and the Health Belief Model

**DOI:** 10.3390/vaccines11020422

**Published:** 2023-02-12

**Authors:** Fahad T. Alsulami, Jesus Sanchez, Silvia E. Rabionet, Ioana Popovici, Mohamed A. Baraka

**Affiliations:** 1Sociobehavioral and Administrative Pharmacy Department, College of Pharmacy, Nova Southeastern University, Fort Lauderdale, FL 33314, USA; 2Clinical Pharmacy Department, College of Pharmacy, Taif University, Taif 21944, Saudi Arabia; 3Clinical Pharmacy Program, College of Pharmacy, Al Ain University, Al Ain P.O. Box 64141, United Arab Emirates; 4Clinical Pharmacy Department, College of Pharmacy, Al-Azhar University, Cairo 11884, Egypt

**Keywords:** HPV, HPV vaccine, human papillomavirus, foreign-born college students, knowledge, acculturation, HBM

## Abstract

Objective: to measure the HPV vaccination rate and knowledge about HPV and its vaccine among foreign-born college students; additionally, to measure the effect of acculturation and HBM constructs on the HPV vaccination behavior among foreign-born college students. Methods: a cross-sectional design with a non-probability sample of foreign-born college students was collected via a web-based self-administered survey that measured the HPV vaccination rate, assessed knowledge about HPV and its vaccine, and evaluated the effect of acculturation and HBM constructs on HPV vaccination behavior among foreign-born college students. Results: Foreign-born college students had moderate knowledge about HPV and the HPV vaccine, and about 63% were HPV-vaccinated. Perceived susceptibility, perceived barriers, and cues to action were significantly associated with the HPV vaccination behavior, while knowledge levels about HPV and the HPV vaccine and acculturation levels were not significantly associated with the HPV vaccination behavior of foreign-born college students. Conclusions: The current study shows a moderate vaccination rate and moderate knowledge about HPV and its vaccine among foreign-born college students. Additionally, vaccination campaigns need to increase awareness about the susceptibility to acquiring HPV and minimize the barriers to receiving the vaccine to increase the HPV vaccination rate among foreign-born college students.

## 1. Introduction

Human papillomavirus (HPV) is the most prevalent sexually transmitted disease (STD), with over 43 million people already carrying HPV and 14 million contracting it yearly in the United States [1]. Furthermore, over thirty thousand HPV-related cancers develop annually in the United States due to HPV infection [2]. The HPV vaccine is the most available approach to prevent most cases of vaginal, vulvar, oropharyngeal, penile, and anal cancer in the United States. Although immunization can start as early as age 9, the Advisory Committee on Immunization Practices (ACIP) advises that all boys and girls receive the HPV vaccine by the age of 11 or 12. Anyone aged 13 to 26 who did not begin or finish the HPV vaccine series by age 12 are also advised to receive a routine catch-up immunization. Healthcare providers may recommend the HPV vaccine for individuals aged 27 to 45, according to a statement from the ACIP [2]. Even though the HPV vaccine has been available for more than ten years and is the most effective way to prevent the virus from spreading, the HPV vaccination rate is below the 80% Healthy People 2020 target [3,4].

In 2018, over 1.7 million foreign-born students attended colleges and universities in the U.S., representing 9% of the U.S. college student population [5]. Regarding the HPV vaccination rate, limited studies measured the HPV vaccination rate among foreign-born college students in the U.S. However, foreign-born people have a lower HPV vaccination rate than U.S.-born individuals. Among 18 to 26-year-old adults, those who were born in the U.S. initiated the HPV vaccine at a rate that was over two times higher than those who were foreign-born individuals. About 16.8% of individuals who were U.S.-born reported completing the HPV vaccine, compared to 7.6% of those who were foreign-born [6]. Language barriers and cultural stigma related to HPV related discussions are common issues associated with the HPV vaccination among foreign-born adults [6].

Foreign-born college students come from countries with different languages, beliefs, values, and backgrounds, and they must choose how much of their cultural identity to preserve and how much to adapt to the host culture when interacting with their new environment [7]. Acculturation is the process of transforming one’s native culture’s social and cultural norms, ideas, beliefs, and behavioral patterns into those of a new society [8]. Acculturation is a latent construct that may be estimated but not precisely measured [8]. In light of this, proxies are carried out to measure the latent construct. Several acculturation scales have been developed to measure acculturation modes [9].

Acculturation significantly impacts how foreign-born adults seek health information [10]. However, few studies measured the effect of acculturation on the HPV vaccination status among foreign-born college students. For instance, foreign-born Chinese college students with significant levels of Western acculturation were more likely to receive the HPV vaccine than those with Asian acculturation [11]. Additionally, Latina mothers with a high level of U.S. acculturation were more likely to vaccinate their daughters against HPV [12]. In contrast, a systematic review by Lara et al. (2005) found that acculturation level had a negative effect on immunization behavior [13]. Therefore, more studies are needed to estimate the effect of acculturation levels on the HPV vaccination status among foreign-born college students.

The Health Belief Model (HBM), a cognitive theory, was developed by psychologists to explain why many individuals failed to participate in a program that prevented and detected disease [14]. The HBM has six constructs: perceived susceptibility, perceived severity, perceived benefits, perceived barriers, self-efficacy, and cues to action [15]. These constructs explain why individuals do not maintain or adopt healthy behaviors. Perceived susceptibility is a person’s perception of their vulnerability to contracting the disease or its consequences. Perceived severity is a person’s perception of the severity or seriousness of the disease or its consequences. A person’s perception of the advantages of engaging in that healthy behavior defines the perceived benefits. Perceived barriers are a person’s perception of whether or not there are obstacles to engaging in that healthy behavior. The confidence that one can perform a recommended activity is known as self-efficacy. Lastly, the cues that could motivate taking a suggested action are known as cues to action [15].

The HBM is one of the most extensively used conceptual frameworks in health behavior research, used to explain changes in health-related behaviors as well as to guide intervention design [15,16]. Numerous health-related behaviors, such as immunization against transmitted infections, have been successfully identified by the HBM [17]. Before this study, no known study measured the effect of the HBM constructs on the HPV vaccination behavior of foreign-born college students in the U.S. However, few studies examined the relationship between HPV vaccination behavior and a limited number of HBM constructs among college students [18,19,20].

While limited studies measured the HPV vaccination rate and knowledge levels about HPV and the HPV vaccine among foreign-born college students in the U.S., this study aimed to measure the HPV vaccination rate and knowledge about HPV and the HPV vaccine among foreign-born college students in the U.S. Additionally, while limited studies measured the effect of acculturation levels of foreign-born college students on HPV vaccination behavior, and no prior research measured the impact of six HBM constructs on the HPV vaccination behavior among foreign-born college students in the U.S., this study aimed to estimate the effect of acculturation levels and the six HBM constructs on the HPV vaccination behavior of foreign-born college students in the U.S.

## 2. Materials and Methods

### 2.1. Participants

A quantitative cross-sectional survey with a non-probability sample of college students at Nova Southeastern University (NSU) was collected via a self-administered web-based survey. REDCap (Research Electronic Data Capture) is a secure online system for developing and administering web-based surveys. In April 2022, an email with a study invitation and survey link was sent to all NSU students. Students were eligible to participate in the study based on these inclusion criteria: (1) 18 years old and older, (2) proficient in English, and (3) able to provide informed consent. To increase the response rate, the first 400 student participants received a USD 15 gift card as compensation for their participation after completing the survey. A total of 2843 college students filled up the online survey. Only foreign-born college students were included in the analysis of this study (n = 376).

### 2.2. Measures

The study survey consisted of a series of validated instruments, including a sociodemographic, HPV vaccination status, knowledge about HPV and the HPV vaccine, HBM constructs, and acculturation instruments. Ten student volunteers pre-tested these instruments to ensure the adequacy of the survey instruments.

#### 2.2.1. HPV Vaccination Behavior

The HPV vaccination behavior was measured by using one item whereby students who received at least one dose of the HPV vaccine were coded as (HPV-vaccinated = 1), and those who did not receive any dose of the HPV vaccine were coded as (non-HPV-vaccinated = 0).

#### 2.2.2. HPV and the HPV Vaccine Knowledge

Students were asked a series of questions regarding HPV and the HPV vaccine (22-item scale) adapted from different studies (e.g., “The human papillomavirus (HPV) vaccination is recommended only for women and girls” “HPV can infect the oral cavity, respiratory tract, and eyes” and “A person usually has symptoms when infected with HPV”) [21,22,23]. Response options were “True”, “False”, or “Don’t Know”. A correct response will be scored as one. Incorrect or do not know responses will be scored as zero. The total knowledge score ranges from 0-to-22. A higher score will indicate a high knowledge level about HPV and the HPV vaccine.

#### 2.2.3. Health Belief Model

The HBM instrument contained six constructs of HBM (perceived susceptibility, perceived severity, perceived benefits, perceived barriers, self-efficacy, and cues to action) adapted from the literature [24,25,26]. The perceived susceptibility construct was measured by using three items (e.g., “If you do not get the human papillomavirus (HPV) vaccine, how likely do you think it is that you will get HPV infection in the future?”). The perceived severity construct was measured using three items (e.g., “How serious an illness do you think HPV is?”). The perceived benefits construct was measured using three items (e.g., “How effective do you think the HPV vaccine is in preventing HPV?”). The perceived barriers construct was measured using four items (e.g., “I would have severe side effects after receiving HPV vaccination”). The self-efficacy construct was measured using two items (e.g., “I am confident that I could take up HPV vaccines if I want to”). Lastly, the cues to action construct was measured by using three items (e.g., “I have watched media reports promoting HPV vaccines”). Student’s response was measured using a 5-point Likert scale; a mean score ranges from 1-to-5, and a higher score indicated students perceived high susceptibility, severity, benefits, barriers, self-efficacy, and cues to action.

#### 2.2.4. Acculturation

Acculturation was measured using a modified version of the East Asian Acculturation Measure (EAAM). EAAM was developed by Berry (2001), contained 29 items, and used a 7-point Likert scale to measure four acculturation modes: Assimilation, Integration, Separation, and Marginalization [9]. Assimilation means a person values the host culture while devaluing their own. Integration means a person likes both the host culture and his or her own culture. Separation means a person prefers their native culture above the host culture. Finally, marginalization means a person dislikes both his or her native and host cultures [9]. Items were modified to refer to all foreign-born college students rather than Asian students only. The student’s response was measured by using a 7-point Likert scale (Strongly disagree = 1, Strongly agree = 7). A mean score of each subscale ranges from 1-to-7. A student was placed either in Assimilation, Integration, Separation, or Marginalization acculturation mode based on his/her highest average score in four acculturation modes. Then, students were classified into two acculturation levels (students in Assimilation or Integration acculturation mode were identified as students with a high acculturation level = 1, and students in Separation or Marginalization acculturation mode were identified as students with a low acculturation level = 0).

### 2.3. Data Analysis

Descriptive analysis was used to summarize and describe the data. The Chi-square test was calculated to test the difference in HPV vaccination rate and acculturation levels across different sociodemographic characteristics of foreign-born college students. Additionally, the independent samples *t*-test was computed to test the mean difference in HPV knowledge scores, perceived susceptibility, perceived severity, perceived benefits, perceived barriers, self-efficacy, and cues to action constructs across different HPV vaccination statutes and acculturation levels. Finally, multivariate logistic regression analysis was conducted to determine the unique effect of independent variables (HPV and the HPV vaccine knowledge levels, HBM constructs, and acculturation levels) on the outcome variable (HPV vaccination status of foreign-born college students). All data analyses were performed using IBM^®^ Statistical Package for the Social Sciences (SPSS), version 28.0.

## 3. Results

### 3.1. Respondent Profile

A total of 376 foreign-born college students were included in this study. About 67% of them were females. The mean age was 25.4 years old, and about 65% were young adult students (18 to 26 years old). About 33% of foreign-born college students identified themselves as international students. Additionally, 42%, 22%, and 20% of foreign-born college students were Hispanic/Latino, White, and Asian, respectively. Further, 58% of foreign-born college students had been living in the U.S for ten years or less. A majority were in a health professional major (69%). Additionally, about 7% were HPV-positive and 5% had a family history of cervical cancer (Table 1).

### 3.2. HPV Vaccination Behavior

Regarding HPV vaccination, about 63% of foreign-born college students received at least one dose of the HPV vaccine. A Chi-square test was performed to identify any significant difference in the HPV vaccination rate across different sociodemographic characteristics. About 71% of female and 46% of male foreign-born college students were HPV-vaccinated (*p*-value < 0.001). Additionally, about 72% and 46% of young adult students (18 to 26 years old) and middle-aged adults (26 years old and older) of foreign-born college students were HPV-vaccinated, respectively (*p*-value < 0.001). Further, 72%, 69%, and 63% of Hispanic/Latino, Black, and White foreign-born college students were HPV-vaccinated, respectively (*p*-value = 0.002). About 71% of foreign-born college students with more than ten years of living in the U.S. were HPV-vaccinated, while about 57% of foreign-born college students with ten years or less of living in the U.S. were HPV-vaccinated (*p*-value = 0.002). About 54% of international foreign-born college students were HPV-vaccinated, while about 68% of non-international foreign-born college students were HPV-vaccinated (*p*-value = 0.009) (Table 2).

### 3.3. Acculturation

According to their acculturation score, 80% of foreign-born college students had a high acculturation level. A Chi-square test was performed to identify any significant difference in the sociodemographic characteristics between the two acculturation levels. About 72% of high acculturated foreign-born college students were female and 25% were international students (*p*-value < 0.001). On the other hand, about 50% of low acculturated foreign-born college students were females and 63% were international students (*p*-value < 0.001). Additionally, about 51% of foreign-born college students with high acculturation had ten years or less of living in the U.S. In comparison, about 85% of foreign-born college students with low acculturation had ten years or less of living in the U.S. (*p*-value < 0.001).

The HPV vaccination rate was higher among foreign-born college students with a high acculturation level than those with a low one. About 66% and 53% of foreign-born college students with high and low acculturation levels received at least one dose of the HPV vaccine, respectively (*p*-value = 0.031) (Table 3).

### 3.4. HPV and the HPV Vaccine Knowledge Levels

Foreign-born college students had moderate knowledge about HPV and the HPV vaccine (M = 12.02 out of 22, SD = 5.7). Eighty-three percent of foreign-born college students did not know that no screening is used to test males for HPV infection. Additionally, about 61%, 60%, and 57% of foreign-born college students did not know that HPV infection can cause penile cancer, anal cancer, and oropharyngeal cancer, respectively. Additionally, about 79% of them did not realize that a condom does not prevent HPV infection (Table 4).

An independent sample *t*-test was conducted to test the mean difference in HPV knowledge levels among foreign-born college students across different HPV vaccination statuses and acculturation levels. Foreign-born college students who received at least one dose of the HPV vaccine had higher knowledge levels about HPV and the HPV vaccine than those who did not receive the HPV vaccine (*p*-value < 0.001). Additionally, foreign-born college students with a high acculturation level had higher knowledge levels about HPV and the HPV vaccine than those with a low acculturation level (*p*-value < 0.001) (Table 5).

### 3.5. HBM Constructs

Foreign-born college students perceived the high severity of HPV and HPV-related diseases (M = 4.07), high self-efficacy to receive the HPV vaccine (M = 4.02), and high benefits of receiving the HPV vaccine (M = 3.99). In addition, they perceived high to moderate susceptibility to acquiring HPV (M = 3.12). Additionally, Foreign-born college students perceived low to moderate cues to receive the HPV vaccine (M = 2.86) and barriers to receiving the HPV vaccine (M = 2.30).

An independent sample *t*-test was conducted to test the mean difference of the six HBM constructs levels among foreign-born college students across different HPV vaccination statuses and acculturation levels. Foreign-born college students who received at least one dose of the HPV vaccine perceived a high susceptibility to acquiring HPV, perceived a high severity of HPV-related diseases, perceived more benefits of receiving the HPV vaccine, perceived fewer barriers to receiving the HPV vaccine, perceived more cues to receive the HPV vaccine, and perceived more self-efficacy to receive the HPV vaccine compared with those who did not receive the HPV vaccine (*p*-value < 0.05).

Additionally, foreign-born college students with high acculturation levels perceived a high severity of HPV-related diseases, perceived more benefits of receiving the HPV vaccine, perceived less barriers to receive the HPV vaccine, and perceived more self-efficacy to receive the HPV vaccine compared with those with low acculturation levels (*p*-value < 0.05) (Table 6).

### 3.6. Predictor of the HPV Vaccination Behavior

Multivariate logistic regression was conducted to determine the effect of independent variables (HPV and the HPV vaccine knowledge levels, HBM constructs, and acculturation levels) on the HPV vaccination behavior of foreign-born college students. The model was significant (χ^2^ = 108.6, *p*-value < 0.001) and explained roughly 34% of the variation in HPV vaccination behavior of foreign-born college students (Nagelkerke *R*^2^ = 0.343). Three constructs of HBM (perceived susceptibility, perceived barriers, and cues to action) were significantly associated with the HPV vaccination status of foreign-born college students. Perceived susceptibility and cues to action were significantly associated with increased HPV vaccination behavior (*p*-value < 0.001). The perceived barrier to receiving the HPV vaccine was significantly associated with decreased HPV vaccination behavior (*p*-value < 0.001). However, knowledge levels about HPV and the HPV vaccine and acculturation levels were not significantly associated with the HPV vaccination uptake of foreign-born college students (Table 7).

## 4. Discussion

This study is one of the few studies that aimed to measure the HPV vaccination rate and HPV and the HPV vaccine knowledge level among foreign-born college students in the U.S. Additionally, it is one of the first studies that aimed to estimate the effect of acculturation levels and the six HBM constructs on the HPV vaccination behavior of foreign-born college students in the U.S.

Foreign-born college students had a moderate HPV vaccination rate. About 63% of foreign-born college students received at least one dose of the HPV vaccine. This HPV vaccination rate is lower than the general college student population in the U.S., where about 75% of college students received at least one dose of the HPV vaccine [27]. Additionally, it is lower than the Healthy People 2020 target of 80% [3]. This finding may be explained by the fact that foreign-born individuals are less likely to have health insurance than U.S.-born individuals [28]. Evidence revealed that individuals with health insurance are more likely to receive preventive therapies such as the vaccine than those without health insurance [29]. Thus, giving foreign-born college students access to free or inexpensive HPV vaccinations could significantly increase the HPV vaccination rate in this population. Additionally, female foreign-born college students had a higher HPV vaccination rate than male foreign-born college students. This finding is consistent with another study, where female college students had a higher HPV vaccination rate than male college students [30]. This finding might be explained by the fact that the HPV vaccine was initially limited to females. The HPV vaccination recommendations and campaigns primarily targeted females, which could affect the HPV vaccination rate among males [31]. However, vaccination of male college students against HPV is crucial because male students report having more risky sexual behavior, such as having two or more sexual partners, compared to their female counterparts [32]. In addition, the findings of this study indicated that foreign-born students over 26 years old had a lower HPV vaccination rate than those 26 years old or younger. This result is consistent with another study [33]. Most people in this age group are not aware that the HPV vaccine has been approved for up to 45 years old, and they do not receive a recommendation to be vaccinated against HPV from healthcare providers [34].

Foreign-born college students with more than ten years length of living in the U.S. had a higher HPV vaccination rate compared with those with ten years or less length of living, and it is consistent with another study [35]. This finding might be explained by the fact that as the U.S. was one of the first countries that provided the HPV vaccine in 2006, foreign-born college students with more than ten years length of living in the U.S. have been exposed to the knowledge related to HPV and the HPV vaccine through public health campaigns on television, the internet, and in printed materials at a younger age that increased their HPV vaccination rate [36].

Foreign-born college students had moderate knowledge about HPV and the HPV vaccine (12.03 out of 22). These results are consistent with other studies, whereas college students have moderate knowledge about HPV and the HPV vaccine [37]. Most foreign-born college students do not know that HPV does not cause genital herpes, and nearly all sexually active individuals will contract HPV at some point in their life. These findings are consistent with other studies [38,39]. In addition, only one-fifth of foreign-born college students knew that condoms do not protect against HPV, consistent with another study [23]. Language barriers and cultural stigma associated with HPV conversations are frequent problems for foreign-born people to increase their knowledge about HPV and the HPV vaccine [6]. Foreign-born college students who received at least one dose of the HPV vaccine had higher knowledge levels about HPV and the HPV vaccine than those who did not receive the vaccine. This finding is consistent with other studies where HPV-vaccinated college students had higher knowledge levels than non-HPV-vaccinated students [38,39]. Thus, increasing the knowledge level about HPV and the HPV vaccine is essential to improving the HPV vaccination rates among foreign-born college students. Additionally, to increase their knowledge levels, vaccination campaigns must target this group with a customized and culturally relevant message delivered in multiple languages. Further, foreign-born college students with low acculturation levels had lower knowledge levels about HPV and the HPV vaccine, consistent with another study [40]. This finding might be explained by the fact that lower English proficiency among foreign-born college students with low acculturation levels is a barrier to improving their health literacy [41].

Perceived susceptibility, perceived barriers, and cues to action constructs of HBM were significant predictors of HPV vaccination behavior among foreign-born college students. Foreign-born college students who perceived high susceptibility to acquiring HPV or HPV-related diseases were more likely to be HPV-vaccinated. This finding is consistent with other studies [20,42]. As a result, to increase the HPV vaccination rate, healthcare providers and vaccination campaigns need to enhance college foreign-born college students’ awareness about the high incidence rate of HPV infection among college-aged individuals. Additionally, they need to inform foreign-born college students that they are at high risk of acquiring HPV, as college students are likely to engage in high risk sexual behaviors. Additionally, foreign-born college students who perceived high barriers to getting the HPV vaccine were less likely to be HPV-vaccinated, and this finding is consistent with another study [19]. Some studies identified some of these barriers. For instance, language barriers, lack of knowledge about HPV and the HPV vaccine, and cultural stigma associated with HPV vacation are frequent problems for foreign-born people [6]. Thus, university student health centers need to minimize these barriers to increase the HPV vaccination rate among foreign-born college students. Lastly, foreign-born college students who perceived more cues of getting the HPV vaccine were more likely to be HPV-vaccinated, and this finding is consistent with other studies [18,42]. However, most college students did not receive a recommendation to receive the HPV vaccine [18]. As a result, healthcare providers play an essential role in the HPV vaccination behavior of foreign-born college students by providing recommendations about receiving the HPV vaccine.

Contrary to expectations, acculturation levels were not significantly associated with the HPV vaccination behavior of foreign-born college students, which is consistent with previous studies. Acculturation level did not affect Latina parents’ decision to vaccinate their children against HPV, and it did not affect the HPV vaccine completion among Latina adolescents [12,43,44]. However, the current study had few participants with low acculturation levels. About 20% of foreign-born college student participants had low acculturation levels. Therefore, more studies are needed to examine the effect of acculturation among large sample size of foreign-born college students.

## 5. Recommendations

The outcomes of this study have significant implications for researchers and healthcare providers. This study indicates that the HPV vaccination rate among foreign-born college students remains lower than the target of 2020 healthy people of 80%. Additionally, the findings of this study highlight a low to moderate knowledge level about HPV and the HPV vaccine among them. The study’s results help identify foreign-born college student categories with lower HPV vaccination rates. Additionally, this study’s results may help develop interventions to encourage foreign-born college students to receive the HPV vaccine. Educating foreign-born college students on the risks and vulnerabilities associated with HPV infection, providing a recommendation for receiving the HPV vaccine, and removing obstacles to receiving the vaccine can enhance their attitudes toward HPV vaccination. Therefore, healthcare professionals and student health centers can provide recommendations and education to encourage more college students to receive the HPV vaccine. Additionally, community pharmacists play a crucial role in boosting the HPV vaccination as they provide immunization services. Community pharmacists help educate foreign-born students about the importance of receiving the HPV vaccine and give recommendations for receiving the vaccine [45]. This study indicates that foreign-born college students with low acculturation levels had lower HPV vaccination rates compared with those with high acculturation levels. Additionally, they had lower knowledge levels about HPV and the HPV vaccine and perceived more barriers to receiving the HPV vaccine than those with high acculturation levels. Thus, healthcare providers including pharmacists need to provide interventions targeting foreign-born college students with low acculturation levels to increase their HPV vaccination behavior and their knowledge levels about HPV and the HPV vaccine and also to minimize their barriers to receiving the vaccine.

## 6. Limitations

Several limitations need to be acknowledged. First, the only language in which the study’s questionnaires were given out was English. Therefore, only foreign-born college students who were proficient in English were able to respond. Second, the convenience sampling method used in this study restricted the generalizability of the results to other populations. Third, foreign-born college students in this study were recruited from one university, which also restricted the generalizability of the results. Fourth, conclusions regarding causality were limited because of the use of a cross-sectional design in this study. Fifth, using a self-report survey in this study may include recall bias, which might overstate or understate the strength of the suggested correlations. Sixth, the lack of standardization for the health belief model instrument for the HPV vaccination behavior is another limitation of this study. Seventh, this study measured HPV and the HPV vaccine knowledge levels without evaluating the awareness of HPV and the HPV vaccine, which might overstate or understate their knowledge levels about HPV and the HPV vaccine. Lastly, this study compared foreign-born college students who received at least one dose of the HPV vaccine and those who did not without estimating the rate of foreign-born college students who received all doses of the HPV vaccine.

## 7. Conclusions

This study was one of few studies that aimed to assess the HPV vaccination rate and knowledge of HPV and the HPV vaccine among foreign-born college students in the U.S. Additionally, it examined the effect of acculturation levels and six HBM constructs on the HPV vaccination behavior of foreign-born college students. This study indicates the percentage of HPV vaccination among foreign-born college students is still below the 80% target for healthy people in 2020. About three-fifths of foreign-born college students received at least one dose of the HPV vaccine, and they had a moderate knowledge level about HPV and the HPV vaccine. In addition, foreign-born college students with low acculturation had very low HPV vaccination rates and very low knowledge levels about HPV and the HPV vaccine compared with those with high acculturation. Perceived susceptibility, perceived barriers, and cues to the action of HBM constructs were significantly associated with HPV vaccination behavior among foreign-born college students. However, acculturation was not significantly associated with HPV vaccination behavior. Therefore, increasing awareness about the vulnerability of acquiring the HPV, recommending the HPV vaccine, and eliminating barriers associated with receiving the vaccine can improve the HPV vaccination rate among foreign-born college students. The results of this study offer relevant findings about HPV vaccination behavior and predictor factors of receiving the HPV vaccine among foreign-born college students in the U.S. Additionally, these findings may help in developing an intervention to enhance the vaccination behavior among this minor group of college students.

## Figures and Tables

**Table 1 vaccines-11-00422-t001:** Sociodemographic characteristics of foreign-born college students.

Sociodemographic Variables	Frequency	Percent (%)
**Age** (mean, SD)	M: 24.42	SD: 5.49
Young Adult Students (18–26)	245	65.2%
Middle-Aged Students (27–45)	131	34.8%
**Gender**		
Female	252	67%
Male	124	33%
**Race/Ethnicity**		
White	84	22.3%
Black	13	3.5%
Hispanic or Latino	159	42.3%
Asian	74	19.7%
Other Ethnicity	46	12.2%
**Length of Living in the U.S.**		
10 Years or Less	218	58%
More Than 10 Years	158	42%
**Study Major**		
Health Professional Major	259	68.9%
Art and Law Major	26	6.9%
Sciences and Engineering Major	38	10.1%
Business and Other Major	53	14.1%
**Degree Level**		
Undergraduate Degree	167	44.4%
Graduate Degree	163	43.4%
Professional Degree	46	12.2%
**Student Status**		
International Student	123	32.7%
Non-International Student	253	67.3%
**HPV Diagnosis**		
Yes	27	7.2%
No	329	87.5%
Don’t Know	20	5.3%
**Family History of Cervical Cancer**		
Yes	18	4.8%
No	324	86.2%
Don’t Know	34	9%
**HPV Vaccination Status**		
HPV-vaccinated	237	63%
Non-HPV-vaccinated	139	37%

**Table 2 vaccines-11-00422-t002:** HPV vaccination rate across different sociodemographic characteristics of foreign-born college students.

Sociodemographic Characteristics	HPV-Vaccinated (n = 237)	Non-HPV-Vaccinated (n = 139)	*p*-Value
**Age**			**<0.001**
Young Adult Students (18–26)	72.2%	27.8%	
Middle-Aged Students (27–45)	45.8%	54.2%	
**Gender**			**<0.001**
Female	71.4%	28.6%	
Male	46%	54%	
**Race/Ethnicity**			**0.002**
White	63.1%	36.9%	
Black	69.2%	30.8%	
Hispanic or Latino	72.3%	27.7%	
Asian	55.4%	44.6%	
Other Ethnicity	41.3%	58.7%	
**Length of Living in the U.S.**			**0.004**
10 Years or Less	56.9%	43.1%	
More Than 10 Years	71.5%	28.5%	
**Study Major**			0.077
Health Professional Major	65.3%	34.7%	
Art and Law Major	76.9%	23.1%	
Sciences and Engineering Major	52.6%	47.4%	
Business and Other Major	52.8%	47.2%	
**Student Status**			**0.009**
International Student	53.7%	46.3%	
Non-International Student	67.6%	26.3%	

**Table 3 vaccines-11-00422-t003:** Characteristics of foreign-born college students based on acculturation level.

Sociodemographic Variables	High Acculturation (n = 298)	Low Acculturation (n = 78)	*p*-Value
**Age**			0.451
Young Adult Students (18–26)	66.1%	61.5%	
Middle-Aged Students (27–45)	33.9%	38.5%	
**Gender**			**<0.001**
Female	71.5%	50%	
Male	28.5%	50%	
**Ethnicity**			**<0.001**
White	23.8%	16.7%	
Black	4%	1.3%	
Hispanic or Latino	45.3%	30.8%	
Asian	19.5%	20.5%	
Other Ethnicity	7.4%	30.8%	
**Length of Living in the U.S.**			**<0.001**
10 Years or Less	51%	84.6%	
More Than 10 Years	49%	15.4%	
**Study Major**			0.560
Health Professional Major	69.1%	67.9%	
Art and Law Major	7%	6.4%	
Sciences and Engineering Major	9.1%	14.1%	
Business and Other Major	14.8%	11.5%	
**Degree Level**			0.655
Undergraduate Degree	43.3%	48.7%	
Graduate Degree	44%	41%	
Professional Degree	12.8%	10.3%	
**Student Status**			**<0.001**
International Student	24.8%	62.8%	
Non-International Student	75.2%	37.2%	
**HPV Vaccination Status**			** 0.031 **
HPV-vaccinated	65.8%	52.6%	
Non-HPV-Vaccinated	34.2%	47.4%	

**Table 4 vaccines-11-00422-t004:** Number of foreign-born college students respondents answering questions about HPV and the HPV vaccine correctly.

HPV and the HPV Vaccine Knowledge Items	Frequency	Percent
1: The human papillomavirus (HPV) vaccination is recommended only for women and girls.	234	62.2%
2: The HPV vaccination is often associated with serious side: effects.	207	55.1%
3: The HPV vaccine is given as a single shot.	264	70.2%
4: It is too late for teenagers who already had sex to have the HPV vaccine.	288	76.6%
5: It is best to receive the HPV vaccine before being sexually active.	243	64.6%
6: HPV infection is a sexual transmitted infection.	297	79%
7: HPV can infect the oral cavity, respiratory tract, and eyes.	195	51.9%
8: HPV infection is related to sexual contact.	270	71.8%
9: Having one type of HPV infection means that you cannot acquire new types.	238	63.3%
10: There is a screening that is commonly used to test males for HPV infection.	376	17%
11: A person usually has symptoms when infected with HPV.	182	48.4%
12: HPV is not a very common virus.	244	64.9%
13: Condoms prevent HPV infection.	78	20.7%
14: HPV infection is related to the development of cervical cancer.	259	68.9%
15: HPV infection can cause genital warts.	237	63%
16: HPV infection can cause genital herpes.	106	28.2%
17: HPV infection can cause penile cancer.	148	39.4%
18: HPV infection can cause anal cancer.	152	40.4%
19: HPV infection can cause oropharyngeal cancer.	163	43.4%
20: Nearly all sexually active men and women will contract HPV infection at some point in their life.	124	33%
21: People can transmit HPV to their partner(s) even if they have no symptoms of HPV infection.	273	72.6%
22: There are many types of HPV.	256	68.1%

**Table 5 vaccines-11-00422-t005:** Mean HPV and the HPV vaccine knowledge score across different HPV vaccination statuses and acculturation levels of foreign-born college students.

Sociodemographic Characteristics	Mean	SD	*p*-Value
**Total Knowledge Score**	12.02	5.7	
**HPV Vaccination Status**			** <0.001 **
HPV-vaccinated	12.83	5.4	
Non-HPV-Vaccinated	10.64	5.9	
**Acculturation Level**			** <0.001 **
High Acculturation	12.95	5.3	
Low Acculturation	8.47	5.5	

**Table 6 vaccines-11-00422-t006:** Mean HBM constructs score across different HPV vaccination statuses and acculturation levels of foreign-born college students, mean (SD).

Sociodemographic Characteristics	Perceived Susceptibility Mean (SD)	Perceived Severity Mean (SD)	Perceived Benefits Mean (SD)	Perceived Barriers Mean (SD)	Cues To Action Mean (SD)	Self-Efficacy Mean (SD)
**HPV Vaccination Status**						
HPV-vaccinated	3.33 (0.88)	4.14 (0.73)	4.09 (0.73)	2.11 (0.89)	3.15 (0.88)	4.17 (0.90)
Non-HPV-Vaccinated	2.76 (0.86)	3.94 (0.76)	3.81 (0.63)	2.63 (0.76)	2.37 (0.81)	3.76 (0.78)
***p*-value**	**<0.001**	**0.014**	**<0.001**	**<0.001**	**<0.001**	**<0.001**
**Acculturation Level**						
High Acculturation	3.14 (0.94)	4.12 (0.72)	4.03 (0.70)	2.16 (0.84)	2.90 (0.95)	4.16 (0.84)
Low Acculturation	3.05 (0.83)	3.85 (0.81)	3.81 (0.73)	2.85 (0.84)	2.70 (0.87)	3.50 (0.85)
***p*-value**	0.436	**0.007**	**0.017**	**<0.001**	0.095	**<0.001**

**Table 7 vaccines-11-00422-t007:** Multivariate logistic regression model of HPV vaccination behavior of foreign-born college students.

Predictor Variables	Coefficient	SE	Wald	OR	CI (95%)	*p*-Value
**HBM**						
Perceived Susceptibility	0.476	0.145	10.95	1.61	(1.21–2.13)	**<0.001**
Perceived Severity	−0.017	0.193	0.007	0.984	(0.67–1.43)	0.931
Perceived Benefits	0.133	0.215	0.385	1.142	(0.75–1.74)	0.535
Perceived Barriers	−0.640	0.187	11.668	0.527	(0.36–0.76)	**<0.001**
Cues To Action	1.011	0.162	39.00	2.749	(2.00–3.77)	**<0.001**
Self-Efficacy	0.236	0.167	1.981	1.266	(0.91–1.75)	0.159
**HPV and the HPV Vaccine Knowledge**	−0.015	0.026	0.330	0.985	(0.93–1.03)	0.566
**Acculturation level**						
High Acculturation	Reference					
Low Acculturation	−0.014	0.318	0.002	0.986	(0.52–1.83)	0.964

## Data Availability

This article contains all of the study’s data.

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
