# Peer review of "Predictor of HPV Vaccination Uptake among Foreign-Born College Students in the U.S.: An Exploration of the Role of Acculturation and the Health Belief Model"

_vaccines, 2023, doi:10.3390/vaccines11020422_

Round 1

Reviewer 1 Report

This is an interesting study but I found the paper difficult to read because of its length and repetitions. Firstly, the introduction should b shortened (please do not divide it into sections). Secondly, I find quite coonfusing the numbers and proportions reported in the different sections of the text. With regard to vaccine coverage among native U.S. and forigners, it seems that it is very low for the complete cycle (17 and 8% respectively). Then the vaccination coverage for one dose appears to be much higher in the study. Thus the problem seems to be the completion of the cycle more than the administration of the first dose. Is that true? if yes, comparing those who got one dose against those who did not would be a limit of the study. Actually, this study does not compare native vs. forigners but foreigners who were or were not vaccinated. Surprisingly, the mean age appears to be rather highIs this represenative of the study population in the USA?

Another limit of the study is that it is conducted on a small population of students recruited by only one university. Thus it is not possible to know to what extent the results may be generalized. 

The Results section is long and the text should be shortened and referred to the tables that are sufficiently clear. The recommendation should be summarized in the conclusions to better address the take home message of the paper.

Reviewer 2 Report

Thank you for the invitation to review this manuscript. I enjoyed reading this paper as the authors have elaborated the objectives and methods very well. The topic is also very important in terms of public health. It is interesting to see that researcher from pharmacy, either social or clinical, are interested in conducting such studies. Indeed, the valuable addition of pharmacy professionals in vaccination program will boost the vaccination coverage and reduce the vaccine hesitancy. I would suggest the authors to provide some information regarding the role of community pharmacists in boosting the vaccination rate for HPV as discussed earlier (https://www.ncbi.nlm.nih.gov/pmc/articles/PMC7753011/).

Please confirm that there were no negative questions in all scales for which reverse coding was opted.

The definition of the foreign-borne students is clear. Please confirm that these are students who are temporarily residing in the USA for their studies and will either settle in the USA or go back to their countries after completing their education. I hope that none of the participants are USA national, even if they were born in some other countries.

What do you mean by HMB in table 7. Do you think that the correlation of constructs of health belief models with vaccination status or intention is more important than logistic regression?

I have no other concern on the results, but I feel that the authors should describe more about the reasons of low knowledge, and vaccination rate among certain demographic features.

Round 2

Reviewer 2 Report

Thank you for revising the manuscript.